# Psychological Impacts and Post-Traumatic Stress Disorder among People under COVID-19 Quarantine and Isolation: A Global Survey

**DOI:** 10.3390/ijerph18115719

**Published:** 2021-05-26

**Authors:** 

**Keywords:** impact of event-scale, PTSD, COVID-19, pandemic, mental health, quarantine, isolation, global survey

## Abstract

Understanding the presence of post-traumatic stress disorder (PTSD) symptoms in quarantined/isolated individuals is essential for decreasing morbidity and mortality caused by the COVID-19 pandemic. However, there is a paucity of evidence quantifying PTSD status globally during confinement in quarantine/isolation facilities during COVID-19. Therefore, we aimed to assess the PTSD status and factors contributing to PTSD development in quarantined/isolated people during pandemic. Using the Impact of Event Scale-Revised (IES-R) scale, our multicentre, multinational, and cross-sectional online survey assessed the psychological impacts on the quarantine/isolation experience of participants suspected or confirmed to have COVID-19, their PTSD status, and various correlates with developing PTSD. We had 944 (35.33%) valid responses (51.1% from females), mostly from Asian countries (635, 71.4%), and 33.9% were healthcare workers. The number of quarantine days in the PTSD symptoms group (using the IES-R cutoff of 24 for symptomatic or full PTSD) was significantly shorter compared to the non-PTSD group (14 (range 14–40) vs. 14 (14–23.75), *p* = 0.031). Lower rates of PTSD symptoms were observed in participants practicing Buddhist religion than in participants having no religion (OR: 0.30; 95% CI: 0.13–0.68; *p* = 0.005); individuals with vocational training had a higher risk of developing PTSD symptoms (OR: 2.28 (1.04–5.15); *p* = 0.043) compared to university graduates. Individuals forced to be quarantined/isolated had higher odds of developing PTSD symptoms than those voluntarily quarantined/isolated (OR: 2.92 (1.84–4.74); *p* < 0.001). We identified several PTSD correlations among individuals quarantined/isolated during the COVID-19 pandemic, including religious practice, reason for quarantine/isolation, education level, and being a case of the infection. These findings can inform worldwide policies to minimize the adverse effects of such social control measures.

## 1. Introduction

The Coronavirus Disease-2019 (COVID-19) pandemic caused by Severe Acute Respiratory Syndrome Coronavirus 2 (SARS-CoV-2) has significantly affected people worldwide. In the absence of definitive treatments and vaccines, health organizations and ministries employed various non-pharmacological interventions to respond to the pandemic situation [1]. Some of them included physical distancing and lockdowns with the isolation of COVID-19 confirmed and suspected patients, quarantine of exposed individuals, travel restrictions, school and workplace closures, cancellation of mass gatherings, rapid testing, proper allocation and use of personal protective equipment, and maintenance of personal hygiene [2,3]. Although these measures partly controlled the COVID-19 outbreak and reduced disease transmission, as evident in China, they caused a negative effect on people’s mental health worldwide [3,4,5,6]. Among the non-pharmaceutical interventions, quarantine and isolation were shown to be the major factors associated with significant psychological impact [7]. The presence of posttraumatic stress disorder (PTSD) symptoms was also noted in quarantined/isolated individuals, healthcare workers (HCWs), and survivors of previous pandemics, including Severe Acute Respiratory Syndrome (SARS) and Middle Eastern Respiratory Syndrome (MERS) [7,8,9,10]. As with COVID-19, PTSD is also associated with excessive activation of the immune system and can induce an inflammatory response [11,12]. This can further increase morbidity and mortality due to COVID-19. Moreover, excessive IL-6 produced on account of the immune response, as seen frequently in patients with COVID-19, has been linked to the development of PTSD [11,13]. A recent narrative review of the mental health effects due to the COVID-19 pandemic also reported the appearance of anxiety and depressive symptoms among healthcare workers worldwide [14]. However, there is a paucity of evidence quantifying PTSD status during confined periods in quarantine/isolation facilities globally during COVID-19. Understanding the presence of PTSD symptoms in quarantined/isolated individuals is necessary for contemplating the psychological impact and decreasing the morbidity and mortality caused by the pandemic worldwide. Information on the experiences of quarantined/isolated people is essential to minimize the adverse effects of such control measures on those people and the society in its entirety [10]. The findings can help government organizations and health ministries worldwide in drafting policies and implementing targeted measures to reduce the development of PTSD in quarantined/isolated people during a pandemic. The findings could also inform the provision of additional support to individuals at an increased risk of adverse psychological effects of quarantine such as PTSD [10]. 

In this study, we aimed to assess the PTSD status of quarantined/isolated people globally during the COVID-19 pandemic and examine the various correlations involved in developing PTSD by using the Impact of Event Scale-Revised (IES-R) scale. This scale is one of the most widely used scales for assessing the symptoms of PTSD in individuals subjected to stress [15]. The scale has good psychometric properties and has been efficiently utilized in different languages worldwide [16,17,18].

## 2. Materials and Methods

### 2.1. Study Design and Participants

This was a multicenter, multinational cross-sectional survey conducted from May to June 2020 to assess the psychological impact of isolation/quarantine experience on individuals suspected or confirmed to have COVID-19. Respondents were both HCWs and non-HCWs at one week to one-month post-discharge from quarantine/isolation. The survey was distributed online by study collaborators using a snowball sampling technique. The collaborators promoted the survey by acting as gatekeepers on various social media platforms. The online survey was also distributed directly to a convenience sample of participants in isolation centers at the time of discharge. 

We reached participants from the following countries in alphabetical order: Afghanistan, Albania, Algeria, Australia, Bahrain, Bangladesh, Bulgaria, Canada, Chile, China, Denmark, Ecuador, Egypt, El Salvador, France, Germany, Greece, Holland, Honduras, Hungary, India, Indonesia, Iraq, Italy, Jordan, Japan, Kazakhstan, Korea, Kuwait, Lebanon, Libya, Malaysia, Mexico, Myanmar, Nepal, Netherlands, New Zealand, Nigeria, Pakistan, Palestine, Philippines, Portugal, Puerto Rico, Qatar, Romania, Russia, Saudi Arabia, Sudan, Syria, Thailand, Timor Leste, Ukraine, United Arab Emirates, United Kingdom, United States, Vietnam, and Yemen. 

### 2.2. Study Questionnaire

The study questionnaire contained 47 questions, including yes/no, open-ended, rating scale, and multiple-choice questions. The questionnaire had three sections. The first section had 15 questions to obtain the sociodemographic characteristics of the participants. The second section, with 10 questions, obtained the quarantine/isolation information of the participants. The third section assessed the psychological impact using the IES-R assessment tool [15,19,20,21]. The IES-R tool has 22 questions, with scores ranging from 0 to 88 points. To assess the participant’s PTSD level, we used the three cutoff thresholds. A score of 0 to 23 was considered as non-PTSD, while a score of 24 to 36 was assessed as having at least a few symptoms of PTSD [19,20]. A score of 37 and above was assessed as PTSD with a score high enough to suppress the immune system even ten years after an impact event [21]. 

The questionnaire was revised by a panel of healthcare professionals that included one psychologist, two epidemiologists, and five physicians. Further validation of the questionnaire was done by a pilot survey of 30 medical students and 5 people who had quarantine experience during the COVID-19 pandemic, and the questionnaire was subsequently modified as required. This validation estimated the time needed to complete the questionnaire and ensured that all the survey questions were phrased clearly and appropriately for comprehension and to avoid bias that might otherwise arise.

Forward and reverse translation of the questionnaire to local languages was performed. The survey, translated by native speakers, was distributed in the following languages: Albanian, Arabic, Bengali, Chinese, English, Filipino, French, German, Hindi, Indonesian, Japanese, Korean, Kurdish, Malay, Malayalam, Nepali, Pashto, Portuguese, Russian, Spanish, Tamil, Thai, Ukrainian, Urdu and Vietnamese. The questionnaire in each language was pretested on three to five native speakers and subsequently validated and modified. 

### 2.3. Statistical Analysis

Statistical analysis was performed to assess the difference between the non-PTSD and PTSD symptoms groups using the *t*-test, chi-square, and Phi and Cramer’s V. Applying the cutoff threshold of 24 points, we classified the disease (symptoms) group and the normal (non-symptoms) group, from which we built a multivariable logistic regression model using the MASS package with a stepwise Akaike information criterion (AIC) method. The analysis was done using R language version 3.6.0 (R Core Team, Vienna, Austria) on Windows 10.

Responses were considered valid when more than 95% of IES-R questions were answered by those quarantined for seven days or more. IP filtering was applied to exclude multiple survey submissions from the same person. Responses containing a missing value in the IES-R questionnaire were filled using multiple imputations by predictive mean matching (PMM).

### 2.4. Ethics

The study was approved by the ethics committee of the School of Tropical Medicine and Global Health, Nagasaki University, Japan (Reference NU_TMGH_2020_117_1). Participation was voluntary, and all participants provided informed consent on the first page of the questionnaire.

## 3. Results

### 3.1. Sociodemographic Characteristics of the Participants

We received 2672 responses from 57 countries, with 944 (35.33%) valid responses and the highest proportion of responses coming from Iraq (9.4%), followed by Bangladesh (9.3%) (Appendix A). Half of the participants were female (51.1%), and the mean age was 29.8 ± 9.61 years, with no age difference between non-PTSD and PTSD symptoms groups (*p* = 0.086). Most of the participants belonged to the Asian continent (71.4%) and the majority were Muslims (50.9%, *n* = 477) (Table 1). 

Most respondents (72.1%) reported their highest education degree to be bachelor’s degree or higher, and 56.7% were employed. Among these participants, 33.9% were HCWs, and 36.9% reported themselves as the family’s main earning member. Regarding the place of COVID-19 exposure, 40.6% of the participants reported hospital or workplace as their source of exposure, and 18.9% could not identify the source, while 12.7% were exposed while traveling. The vast majority (70%) reported to be in quarantine/isolation voluntarily (Table 1). 

### 3.2. Factors Associated with PTSD Symptoms

The number of quarantine days in the PTSD symptoms group was significantly shorter than that in the non PTSD group (14 (range 14–40) vs. 14 (range 14–23.75), *p* = 0.031). The marital status, the average income, the reason for quarantine/isolation, the place of isolation, whether it was voluntary to be quarantine/isolated, and the participants’ continent were significantly different between the non-PTSD and PTSD symptoms groups (Table 1).

The divorced/widowed/separated participants group had higher PTSD incidence than the married/domestic partnership group which in turn had higher incidence than the single group.

Higher income groups reported more PTSD, with the above 1000 USD group on the top, followed by the 750–1000 group, while the lower income groups reported less PTSDs than non-PTSD.

PTSD occurrence was also different between continents. In Africa, America, Oceania, and Europe, the PTSD group was higher than the non-PTSD group, while it was the reverse in Asia.

Those who were quarantined because they were confirmed to have COVID-19 (F0) reported more PTSD than non-PTSD. This trend was similar in participants who had close contact with confirmed patients (F1); F2, who had close contact with people who had a direct or indirect contact with a confirmed case; F3, who had close contact with F2; F4, who had close contact with F3. Non-PTSD was greater than PTSD cases in participants who were quarantined because they lived, stayed, or worked at a place nearby a confirmed COVID-19 patients group, and those who returned from affected geographical areas group.

### 3.3. Predictive Factors of PTSD Symptoms

Table 2 illustrates the explanatory variables in the multivariate ordinal regression model that contributed to predicting the presence of PTSD symptoms. In the regression results, there was a decreased likelihood of PTSD symptoms when an individual belonged to the Buddhist religion compared to no religion (OR: 0.30; 95% CI: 0.13–0.68; *p* = 0.005). Individuals who possessed vocational training had a higher risk of developing PTSD symptoms (OR: 2.28; 95% CI: 1.04–5.15; *p* = 0.043) than those with a higher education level (Master/Ph.D./Doctoral). Isolated participants who were in the F0 group had a significantly increased risk of developing PTSD symptoms compared to those quarantined due to other reasons (except the F1 individuals). The study also found that individuals who were forced to be quarantined/isolated had higher odds of developing PTSD symptoms than those voluntarily quarantined/isolated (OR: 2.92; 95% CI: 1.84–4.74; *p* < 0.001).

## 4. Discussion

Our survey examined some of the critical factors regarding the quarantine and isolation experience during the COVID-19 pandemic contributing to the development of PTSD symptoms in affected people. Our results revealed that Buddhists were less likely to experience PTSD symptoms compared to those with no religion. This accords with other previous studies that religious faith positively impacts mental health and may help people cope with emotional problems [22]. Buddhism was reported to be indirectly involved with both social adaptation status and psychological well-being [23]. Also, our study showed that individuals who possessed vocational training had a higher risk of developing PTSD symptoms than those with higher education levels (university graduates). A better perception could explain this, regarding the risk of infection during quarantine or isolation period among those with higher education level; therefore, these participants had a lower risk of PTSD. 

The reason for being quarantined or isolated appeared to be one of the major indicators of the IES-R score, with individuals who had confirmed cases of the infection (F0) scoring higher on the IES-R scale compared to those who were not in contact with a confirmed case (F2, F3, F4, living nearby, or returning from an affected geographical area). Similar findings were demonstrated in several other studies conducted during the present COVID-19 pandemic and the previous outbreaks. Mak et al. reported the prevalence of PTSD in one-fourth of SARS survivors 30 months after illness, making it the most prevalent psychiatric disorder among these individuals [24]. Similarly, in a study in MERS survivors, Lee et al. reported chronic fatigue symptoms that presented 12 months following the infection, which contributed to prolonged PTSD symptoms at 18 months [9]. Recently, among 402 COVID-19 survivors screened one month post-discharge, 28% reported PTSD and 56% presented with at least one of the clinical dimensions of PTSD, anxiety, depression, obsessive-compulsive symptoms, and insomnia [25]. During the SARS outbreak, hospital employees who had been quarantined, who had friends or close relatives that contracted the infection, and who had worked in high-risk wards, were two to three times more likely to experience PTSD compared to those not exposed [26]. 

Our study showed that willingness to quarantine or isolate was also related to the IES-R scores, with individuals who were forced into quarantine/isolation scored higher than those who quarantined voluntarily. This is consistent with previous studies that link individual acceptance to psychological impact. The relationship between the outbreak and PTSD and depressive symptoms in 549 hospital employees was evaluated three years after the SARS outbreak [27]. The authors reported that individuals who had an altruistic acceptance of the situation/infection risk had a lower likelihood of experiencing depressive symptoms and PTSD compared to those who could not accept it [26,27]. There have not been any studies directly comparing the psychological effects of mandatory versus voluntary quarantine. However, our study findings are in line with the current data regarding the altruistic acceptance of quarantined or isolated people from previous outbreaks. 

Another factor that appeared to affect IES-R scores was the place of quarantine or isolation. Being quarantined at home was associated with slightly higher IES-R scores than being quarantined in a government-designated center, though not significantly. Although no data examined the association between the place of quarantine and PTSD specifically, in a study on 1800 HCWs and 73 hemodialysis patients during the MERS outbreak, the hemodialysis patients had lower levels of anxiety and depression in hospital compared to home-quarantined HCWs [8]. While this is not consistent with our findings, it is prudent to note that the number of hemodialysis patients and HCWs included in this study was significantly disproportionate, which may be a confounder. Similarly, only 5% of the participants in our study reported to be quarantined in a hospital setting, while 70% were home quarantined, which likely skewed our analysis. Further studies examining the relationship between quarantine and its psychological effects are required to reach a more definitive conclusion.

Our present study did not identify any differences in IES-R scores between HCWs and non-HCWs. This is contrary to some previous findings. Higher PTSD scores among HCWs than the general public were found associated with their high-risk work setting [8]. Nevertheless, our findings are also similar to previous studies. In Canada, during the SARS outbreak, Hawryluck et al. observed that the HCWs status did not affect PTSD symptoms among quarantined individuals [10]. Similarly, Lebanon’s study also reported no significant differences in PTSD symptoms between HCWs and non-HCWs who were quarantined during the COVID-19 pandemic [28]. 

Our study is not without limitations. Despite receiving responses from 57 countries, the number of responses from each country and continent were not equally distributed. There is a possibility that the preventive measures applied in each of these countries, particularly the over-represented regions, might have affected the overall analysis. We also did not conduct an in-depth interview which would give more qualitative insight, and hence, we cannot not completely discount the possibility of some degree of background consequences resulting in similar experiences [29]. Moreover, since we did not include questions regarding various infection-control measures implemented in the participating countries, we could not comment on the extent of their influence on the study results. Also, due to the cross-sectional context of the survey on the quarantine subjects, we could not collect reliable data regarding the subjects’ pre-quarantine psychological state to adjust for the baseline psychological status of the participants as a covariate in the comparison of the non-PTSD and PTSD symptoms groups. The cross-sectional design is not considered the most accurate method of data sampling due to the risk of reporting bias since the participants who finished the questionnaires might be in the circle of acquaintance of the local collaborators, which could reduce the study’s representation. However, the method employed in this study has several benefits, including minimizing the risk of COVID-19 community transmission, accessibility to participants in COVID-19 isolated/quarantined areas, and also compliance with various countries’ social distancing policy or recourse-scarce situations. To improve the validity of survey findings, significant effort was made through piloting and carefully revising the online questionnaire before the actual data collection stage. We also collected data on a global scale, which further improved the divergence of participants in our study.

## 5. Conclusions

Our multicentre, multinational cross-sectional survey has identified several key factors correlating with the development of PTSD symptoms among quarantined/isolated individuals during the COVID-19 pandemic. Religious practice, reason for being quarantined/isolated, education level, and having a positive diagnosis of the infection (F0) remain the predictive factors of stress during pandemics. These findings may inform government organizations and health ministries worldwide in drafting policies and implementing targeted measures to prevent the development of PTSD in quarantined/isolated people during a pandemic, and to reduce morbidity and mortality.

## Figures and Tables

**Table 1 ijerph-18-05719-t001:** Sociodemographic profile of participants and their descriptive summary by non-PTSD and PTSDs groups.

	Non-PTSD	PTSDs	Total	*p*-Value
	*N* = 284	*N* = 660	*N* = 944	
**Age (year) (*n* = 931)** [Mean, SD]	29.0 (9.28)	30.1 (9.73)	29.8 (9.61)	0.086
**Number of quarantine days (*n* = 812)** [Median (IQR)]	14.0 (14.0; 40.0)	14.0 (14.0; 23.8)	14.0 (14.0; 30.0)	0.031
**IES-R scores**				
IES-R INT score^a^ [Mean (SD)]	0.48 (0.35)	1.65 (0.62)	1.30 (0.77)	<0.001
IES-R AVO score ^a^ [Mean (SD)]	0.64 (0.46)	1.91 (0.64)	1.53 (0.83)	<0.001
IES-R HYP score ^a^ [Mean (SD)]	0.53 (0.41)	1.90 (0.81)	1.49 (0.95)	<0.001
IES-R score ^a^ [Mean (SD)]	0.53 (0.30)	1.75 (0.53)	1.38 (0.73)	<0.001
**Gender (*n* = 933)**				0.552
Female	139 (49.5)	338 (51.8)	477 (51.1)	
Male	142 (50.5)	314 (48.2)	456 (48.9)	
**Race (*n* = 932)**				0.056
White/Caucasian	44 (15.8)	118 (18.1)	162 (17.4)	
Asian	172 (61.6)	343 (52.5)	515 (55.3)	
Hispanic/Latino	22 (7.9)	79 (12.1)	101 (10.8)	
Others	41 (14.7	113 (17.3)	154 (16.5)	
**Religion (*n* = 937)**				<0.001
No religion	28 (10.0)	61 (9.30)	89 (9.50)	
Buddhist	46 (16.4)	26 (4.0)	72 (7.7)	
Christian ^b^	57 (20.3)	151 (23.0)	208 (22.2)	
Hindu	20 (7.1)	60 (9.2)	80 (8.5)	
Muslim	126 (44.8)	351 (53.5)	477 (50.9)	
Others	4 (1.4)	7 (1.1)	11 (1.2)	
**Marital Status (*n* = 935)**				0.015
Single	178 (63.6)	366 (55.9)	544 (58.2)	
Divorced/Widowed/Separated	5 (1.8)	34 (5.2)	39 (4.2)	
Married/Domestic partnership	97 (34.6)	255 (38.9)	352 (37.6)	
**Level of education (*n* = 937)**				0.126
Master/PhD/Doctoral	44 (15.6)	99 (15.1)	143 (15.3)	
University (undergraduate)	146 (51.8)	318 (48.5)	464 (49.5)	
Vocational training	23 (8.16)	92 (14.0)	115 (12.3)	
Primary school/Secondary school/High school	68 (24.1)	144 (22.0)	212 (22.6)	
No formal education	1 (0.4)	2 (0.3)	3 (0.3)	
**Employment status (*n* = 939)**				0.612
Full time employment	107 (37.8)	236 (36.0)	343 (36.5)	
Casual employment	17 (6.0)	47 (7.2)	64 (6.8)	
Part-time employment	30 (10.)	95 (14.5)	125 (13.3)	
Retired	25 (8.8)	70 (10.7)	14 (1.5)	
Student	4 (1.4)	10 (1.5)	263 (28.0)	
Unemployed	88 (31.1)	175 (26.7)	95 (10.1)	
Others	12 (4.2)	23 (3.5)	35 (3.7)	
**Average income (USD/month) (*n* = 884)**				0.002
<250	120 (44.6)	205 (33.3)	325 (36.8)	
250–500	50 (18.6)	100 (16.3)	150 (17.0)	
500–750	34 (12.6)	84 (13.7)	118 (13.3)	
750–1000	26 (9.67)	79 (12.8)	105 (11.9)	
Over 1000	39 (14.5)	147 (23.9)	186 (21.0)	
**Main laborer in the family (*n* = 922)**				0.316
No	184 (65.7)	398 (62.0)	582 (63.1)	
Yes	96 (34.3)	244 (38.0)	340 (36.9)	
**Healthcare worker (*n* = 930)**				0.107
No	197 (70.1)	418 (64.4)	615 (66.1)	
Yes	84 (29.9)	231 (35.6)	315 (33.9)	
**Reason for quarantine/isolation (*n* = 892)**				<0.001
F0	14 (5.3)	111 (17.6)	125 (14.0)	
F1	27 (10.3)	112 (17.8)	139 (15.6)	
F2/F3/F4	43 (16.4)	70 (11.1)	113 (12.7)	
I live, stay or work at a place nearby a confirmed COVID-19 patient	55 (21.0)	101 (16.0)	156 (17.5)	
I returned from affected geographic areas	54 (20.6)	102 (16.2)	156 (17.5)	
Others	69 (26.3)	134 (21.3)	203 (22.8)	
**Place of exposure (*n* = 855)**				0.129
In hospital	61 (24.1)	137 (22.8)	198 (23.2)	
At home	18 (7.1)	54 (9.0)	72 (8.4)	
At hotel/At a hall, concert, cinema	10 (4.0)	27 (4.5)	37 (4.3)	
At workplace	33 (13.0)	116 (19.3)	149 (17.4)	
During travel by airplane/by bus/by taxi/by train	37 (14.6)	72 (12.0)	109 (12.7)	
I do not know the source of my exposure	46 (18.2)	116 (19.3)	162 (18.9)	
Others	48 (19.0)	80 (13.3)	128 (15.0)	
**Place of isolation (*n* = 914)**				0.008
At home	203 (73.0)	465 (73.1)	668 (73.1)	
At the designated place by the Government	69 (24.8)	128 (20.1)	197 (21.6)	
In hospital	6 (2.2)	43 (6.8)	49 (5.4)	
**Which of the following was true about your quarantine/isolation? (*n* = 911)**				<0.001
I was forced to quarantine/isolate	59 (21.6)	215 (33.7)	274 (30.1)	
I voluntarily quarantined/isolated	214 (78.4)	423 (66.3)	637 (69.9)	
**Who are you quarantined with (*n* = 910)**				0.741
No one else, only me	175 (65.1)	426 (66.5)	601 (66.0)	
Family/friends/colleagues	94 (34.9)	215 (33.5)	309 (34.0)	
**Comfortable in isolation time (*n* = 920)**				0.491
Not at all	15 (5.4)	50 (7.8)	65 (7.1)	
A little bit	38 (13.7)	92 (14.3)	130 (14.1)	
Moderately	112 (40.3)	235 (36.6)	347 (37.7)	
Quite a bit	67 (24.1)	172 (26.8)	239 (26.0)	
Extremely	46 (16.5)	93 (14.5)	139 (15.1)	
**Continent (*n* = 932)**				0.002
Africa	18 (6.4)	70 (10.7)	88 (9.4)	
America	26 (9.3)	99 (15.2)	125 (13.4)	
Asia	201 (71.8)	434 (66.6)	635 (68.1)	
Oceania	0 (0.0)	2 (0.3)	2 (0.2)	
Europe	35 (12.5)	47 (7.2)	82 (8.8)	

Figures in the parentheses indicate percentage (%), unless stated otherwise. Cutoff point was considered at 24. Non-PTSD and PTSDs groups were compared by Chi-square test and Student’s *t*-test for categorical and continuous variables, respectively. *p* < 0.05 was considered significant. PTSD, post-traumatic stress disorder; SD, standard deviation; IES-R INT, Impact of Event Scale Revised Intrusion Symptomatology; IES-R AVO, Impact of Event Scale Revised Avoidance Symptomatology; IES-R HYP, Impact of Event Scale Revised Hyperarousal Symptomatology; IES-R, Impact of Event Scale Revised; ^a^ Scores expressed as average score per question (*n* = 944); ^b^ Christian (including Church of England, Catholic, Protestant and all other Christian denominations).

**Table 2 ijerph-18-05719-t002:** Multivariable regression analysis.

Variables	Univariate Analysis		Multivariable Analysis	
	OR (95% CI)	*p*-Value	OR (95% CI)	*p*-Value
**Religion**				
No religion	Reference		Reference	
Buddhist	0.22 (0.10–0.45)	<0.001	0.30 (0.13–0.68)	0.005
Christian ^a^	1.12 (0.58–2.11)	0.735	0.97 (0.47–1.95)	0.924
Hindu	2.57 (1.01–7.18)	0.056	2.74 (0.98–8.32)	0.062
Muslim	1.26 (0.68–2.29)	0.448	1.68 (0.85–3.27)	0.130
Others	0.82 (0.19–4.20)	0.792	1.32 (0.28–7.44)	0.732
**Level of education**				
Master/PhD/Doctoral	Reference		Reference	
University (undergraduate)	0.88 (0.53–1.42)	0.599	0.79 (0.45–1.35)	0.396
Vocational training	2.12 (1.07–4.34)	0.035	2.28 (1.04–5.15)	0.043
Primary school/Secondary school/High school	0.79 (0.45–1.37)	0.407	0.84 (0.44–1.57)	0.581
**Reason for quarantine/isolation**				
I was F0	Reference		Reference	
I was F1	0.60 (0.27–1.28)	0.193	0.62 (0.27–1.35)	0.231
I was F2/F3/F4	0.24 (0.11–0.48)	<0.001	0.35 (0.15–0.76)	0.010
I live, stay or work at a place nearby a confirmed COVID-19 patient	0.21 (0.10–0.41)	<0.001	0.27 (0.12–0.57)	0.001
I returned from affected geographic areas	0.25 (0.12–0.49)	<0.001	0.39 (0.18–0.83)	0.017
Others	0.24 (0.11–0.47)	<0.001	0.31 (0.14–0.64)	0.002
**Place of isolation**				
At the designated place by the Government	Reference		Reference	
At home	1.44 (0.98–2.08)	0.158	1.61 (0.99–2.59)	0.054
In hospital	3.38 (1.35–10.31)	0.016	2.05 (0.71–6.98)	0.211
**Which of the following was true about your quarantine/isolation?**				
I voluntarily quarantined/isolated	Reference		Reference	
I was forced to quarantine/isolated	1.98 (1.35–2.96)	0.001	2.92 (1.84–4.74)	<0.001
**Main laborer in the family**				
No	Reference		Reference	
Yes	1.31 (0.93–1.86)	0.131	1.43 (0.94–2.19)	0.095

^a^ Christian (including Church of England, Catholic, Protestant and all other Christian denominations). Cutoff point was considered at 24. Multivariable regression analysis was performed to find out the independent predictor of developing PTSD. *p* < 0.05 was considered significant. PTSD, post-traumatic stress disorder; OR, odds ratio; CI, confidence interval.

## Data Availability

The data that support the findings of this study are available from the corresponding author upon reasonable request. Major data is included in the manuscript and Appendix A.

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
