# Peer review of "Psychological Impacts and Post-Traumatic Stress Disorder among People under COVID-19 Quarantine and Isolation: A Global Survey"

_ijerph, 2021, doi:10.3390/ijerph18115719_

Round 1

Reviewer 1 Report

The study design is highly problematic, especially as the data was collected via snowball and convenience sampling, during a very brief period and in an (uncontrolled) self-report manner. 

Specific comments:

  1. Abbreviations such as "SARS-CoV-2" must be properly defined in the first instance of its use.
  2. Instead of citing references [8] and [9] which pertain to MERS, suggest authors cite a more recent review on the impact of the pandemic on healthcare workers (citation: pubmed.ncbi.nlm.nih.gov/32603985) in the introduction section.
  3. Did the authors take steps to guard against duplicate responses to your survey? E.g. IP filtering?
  4. Did the study obtain prior institutional review board (IRB) approval? Please specify the IRB study/approval number.
  5. Were all the countries sampled under lockdown/quaratine during the study period? There is no universal definition for what constitutes a lockdown. They can include full- or partial closures of different types of facilities, such as schools or workplaces.
  6. The average income is per day? Please specify. Also, would it be sensible or fair to compare the different income brackets in different countries given the vastly different standards and costs of living?
  7. Rather than over-pathologize these experiences, some degree of sadness, anxiety, fear, anger, paranoia, and short-term adjustment issues and long-term adaptation to the uncertain future are perhaps reasonable or expected responses. The majority of mental disorders following COVID-19 may be “reactive” in nature. It may be in response to the fear and stress of contagion, especially given the possibility of asymptomatic spreaders in the community. It may be the consequence of hospitalization for infected individuals or strict measures to curb and contain the pandemic, with “lockdown” living, loss of livelihood, and financial hardship (citation: pubmed.ncbi.nlm.nih.gov/32943541). It is clear that this pandemic has disproportionately impacted racial minorities and lower-income families (citation: pubmed.ncbi.nlm.nih.gov/32391864).
  8. Did the authors adjust for baseline depression or anxiety as a covariate?
  9. The issue of selection bias, which is a major study limitation, was not adequately discussed.
  10. The underlying data should be made publicly available. If this was not possible, please provide a reason why.

Author Response

Reviewer’s comments and response

Reviewer#1

General Comment: The study design is highly problematic, especially as the data was collected via snowball and convenience sampling, during a very brief period and in an (uncontrolled) self-report manner. 

Response: Thank you for your time to review our manuscript. Although recognizing that snowball and convenience sampling might contain certain types of risk, this study design helped us reaching to the participants under strict quarantine/isolation during the beginning of the COVID-19 pandemic. This study population was not easy to reach and considered a hard-to-survey population; that too for a multi-centric survey. Therefore, we applied the snowball method to distribute questionnaires through social networks and collaborators in several countries to recruit participants, which also had an advantage of expanding the geographical scope of the study and diversifying the participation. Besides this, snowball and convenience sampling techniques can provide a relatively trustful way of researching with a minimum risk of COVID-19 transmission in the community, which was our top priority for participants and researchers/collaborators during that time. Meanwhile, we could still get access to the study participants regardless of their strictly isolated areas.

Regarding the brief period conducting the survey, we conducted data collection with a multi-national group of local experienced collaborators who also had led the similar studies by using the snowball sampling previously in many studies. In fact, we had already optimized the study process for e.g. preparation of protocols, effectively managing local collaborators and participants, and quality data collecting. Therefore, we believe that our approach minimizes the risk of collecting low-quality data while still achieving adequate responses for the study.

Specific comments:

  1. Abbreviations such as "SARS-CoV-2" must be properly defined in the first instance of its use.

Response: Thank you for pointing it out. We have defined it at its first appearance in the revised manuscript. (Page-2, Line 27- 28)

  1. Instead of citing references [8] and [9] which pertain to MERS, suggest authors cite a more recent review on the impact of the pandemic on healthcare workers (citation: pubmed.ncbi.nlm.nih.gov/32603985) in the introduction section.

Response: Thank you for your valid suggestion. As suggested, we cited a more recent review on the impact of the pandemic on healthcare workers (citation: pubmed.ncbi.nlm.nih.gov/32603985) in the introduction section of revised manuscript. (Page-2, Lines 45-47)

While we still keep the MERS related reference in the manuscript, intentionally, we want the readers recall the similarity of previous historical events and provide an idea of a bigger/ broader picture on this topic.

  1. Did the authors take steps to guard against duplicate responses to your survey? E.g. IP filtering?

Response: Yes, we used IP filtering to prevent the duplicate response. We have, now, mentioned it in the methods section (Page-4, line 5-6).

  1. Did the study obtain prior institutional review board (IRB) approval? Please specify the IRB study/approval number.

Response: Yes. The study protocol was reviewed by IRB and this has been mentioned in the manuscript just before the References. In the revised version, we have included it in the Methods section too (Page-4, Lines 8-11) which t reads as:

“Ethics

The study was approved by the ethics committee of the School of Tropical Medicine and Global Health, Nagasaki University, Japan (Ref. NU_TMGH_2020_117_1). Participation was voluntary, and all participants provided informed consent on the first page of the questionnaire.”

  1. Were all the countries sampled under lockdown/quarantine during the study period? There is no universal definition for what constitutes a lockdown. They can include full- or partial closures of different types of facilities, such as schools or workplaces.

Response: Thank you for your query. Our study only recruited the participants who experienced the quarantined/isolated period. This means that we recruited only participants who had undergone the period of quarantine or isolation at home, in hospitals or at designated quarantine centers/ places. This does not necessarily confirm that the countries sampled were under lockdown at the time of this study.

  1. The average income is per day? Please specify. Also, would it be sensible or fair to compare the different income brackets in different countries given the vastly different standards and costs of living?

Response: To be specific, the question was about the “average monthly income” of the participant in the previous year. We agree that the monthly income may not perfectly illustrate our study participants, but by far, we consider them as the suitable one for this study.

We also agree and understand that the question might be somewhat sensitive for a group of particular participants to answer. Having said that, we considered this question as a ‘must-not-miss’ one in our survey questionnaire because the income of the participant is one of the significant resources/variable to be assessed when one faces with stressful situations. Participants’ monthly income until now would be a better variable for measurement of the quality of life, instead of using the specific national GDP or other factors which are widely considered to be general for a multi-national study with a diversity of people (different occupational and social status) and may not represent such study population.

  1. Rather than over-pathologize these experiences, some degree of sadness, anxiety, fear, anger, paranoia, and short-term adjustment issues and long-term adaptation to the uncertain future are perhaps reasonable or expected responses. The majority of mental disorders following COVID-19 may be “reactive” in nature. It may be in response to the fear and stress of contagion, especially given the possibility of asymptomatic spreaders in the community. It may be the consequence of hospitalization for infected individuals or strict measures to curb and contain the pandemic, with “lockdown” living, loss of livelihood, and financial hardship (citation: pubmed.ncbi.nlm.nih.gov/32943541). It is clear that this pandemic has disproportionately impacted racial minorities and lower-income families (citation: pubmed.ncbi.nlm.nih.gov/32391864).

Response: Thank you for your logical opinion and valid suggestions. We agree and appreciate that there is a gray area between the typical response and the pathological malfunction or disorder, especially during COVID-19, which is a global crisis impacting health care workers, patients, and isolated/quarantined population. We used the IES-R score to set up a cut-off point. Distinguishing between reactions could be considered a normal reaction to environmental change, which is regarded as a disorder or pathological malfunction.  Nevertheless, we cannot not discount the possibility of some level of background consequences.

Although every method has its limitations, the IES-R score with the cut-off point could provide insight to the two groups of participants. While the normal group (e.g. non-PTSD) with the reaction to these traumas or changes could be self-limited, the group with a higher IES-R score would need more specific care and support in various ways of mental, physical, or medical treatment to reduce the severity of their symptoms.

We have also cited these literatures to indicate the potential background situations (in our limitations). (Pge-6, Line 9-10)

  1. Did the authors adjust for baseline depression or anxiety as a covariate?

Response: We are grateful for pointing out this limitation of our study. Due to the cross-sectional context of the survey among the quarantined/isolated subjects, we could not collect the reliable data regarding the pre-quarantine psychological state to adjust for baseline psychological status of the participants as a covariate in the comparison of the non-PTSDs and PTSDs groups. Now, we have added this point while addressing the study's limitations in the revised manuscript.  (Page-6, Lines 13-16)

  1. The issue of selection bias, which is a major study limitation, was not adequately discussed.

Response: Thank you for your respectful suggestion. We have added more details in the discussion part regarding this issue.  (Limitations: Page-6, Lines 17-25)

“Our study is not without limitations. Despite receiving responses from 57 countries, the number of responses from each country and continent was not equally distributed. There was a possibility that the preventive measures applied in each of these countries, particularly the over-represented regions, might affect the overall analysis. Moreover, we did not include the questions regarding various infection-control measures implemented in the participating countries, we could not comment on the extent of their influence on the study results.”

We have also added more limitations in the revised manuscript as suggested, which reads as- “Also, due to the cross-sectional context of the survey on the quarantine subjects, we could not collect the reliable data regarding the pre-quarantine psychological state to adjust for baseline psychological status of the participants as a covariate in the comparison of the non-PTSDs and PTSDs groups. The cross-sectional design is not considered the most accurate of data sampling due to the risk of reporting bias since the participants who finished the questionnaires might be in the circle of acquaintance with the local collaborators, which could reduce the study representation. However, the method employed in this study has several benefits, including minimizing the risk of COVID-19 community transmission, accessible for participants in COVID-19 isolated/quarantined areas, and also in compliance with various countries’ social distancing policy or recourse-scarce situations. To improve the validity of survey findings, significant effort was made through piloting and carefully revising the online questionnaire before the actual data collection stage. We also collected data on a global scale, which further improved the divergence of participants in our study.”

  1. The underlying data should be made publicly available. If this was not possible, please provide a reason why.

Response: Thank you for your suggestion. As our study was a multicenter, multinational cross-sectional survey, while recognizing that the public disclosure of these data may help countries adjust the quarantine/isolation policies to minimize the psychological impact on quarantined/isolated people, the publication of underlying data online may reveal some delicate information regarding quarantine and isolation policies in each country. Therefore, we have decided not to make these data publicly available. However, upon reasonable request, the corresponding author of this manuscript shall provide the appropriate data (completely de-identified) to the researchers/reader.

Reviewer 2 Report

This is a timely survey of background factors which can be predictive as to who will develop PTSD as a reaction to being in quarantine/isolation due to the Covid-19 pandemic. The statistical analysis is useful. However, the article has limitations expressed by the authors (e.g. lack of equal distribution of repsonses from different countries/backgrounds), but they did not consider the need to have a more in-depth investigation via interviews of a smaller sample. It seems to me that it is jsutified to acknowledge this signiciant limitation to our understanding of the responses people have to being isolated within the covid-19 context.

Author Response

Reviewer#2

This is a timely survey of background factors which can be predictive as to who will develop PTSD as a reaction to being in quarantine/isolation due to the Covid-19 pandemic. Statistical analysis is useful. However, the article has limitations expressed by the authors (e.g. lack of equal distribution of responses from different countries/backgrounds), but they did not consider the need to have a more in-depth investigation via interviews of a smaller sample. It seems to me that it is justified to acknowledge this significant limitation to our understanding of the responses people have to being isolated within the covid-19 context.

Response: Thank you for your time to review our manuscript, and a constructive feedback and appreciation. Since we did not perform more investigations, like in-depth interview, we have now included this as one of our limitations in the revised manuscript as suggested. (Page-6, Lines 8-10)

Reviewer 3 Report

Comments: The author examined associations between socio-demographics and PTSD in several countries. The international scope of the study is impressive.

Abstract

-Please define the first occurrence of the acronym IES-R

-The final sentence “Religion, reason for quarantine/iso- 44 lation, education level, and being cases of the infection were major predictive factors of stress during the pandemic” is a bit inaccurate; stress is different from clinical PTSD. Consider adding the list of reasons to the previous sentence or deleting e.g., “, including religion, reason. . .etc.)

Methods

-Why did the authors use a threshold of 24 if the PTSD diagnostic threshold is 37? They should not describe the participants as having PTSD if they do not meet diagnostic criteria. They should instead state that these individuals had elevated PTSD symptoms, or some other non-diagnostic term of their choosing.

-How did the authors choose which variables to include in their regression model? Ideally there should be a quantitative, theoretical, or evidence-based reason for each variable in the model, i.e., a justification for positing an association between the variable and PTSD. Otherwise, we run into the “kitchen sink regression” problem, where numerous independent variables are selected without proper justification.

Was the study evaluated by an institutional review board or other administrative body tasked with human subjects protection?

Results

  • “The marital status, the average income, 149 the reason for quarantine/isolation, the place of isolation, the voluntary to be quarantine/isolated, 150 and the participants’ continent were significantly different between the normal and PTSD groups.(p.3, line 149)” Please specify the direction of the difference between groups for each variable . Higher, lower?
  • I would recommend using a term such as “No PTSD”, “Non-symptomatic” etc. instead of “Normal.” It implies that people with PTSD are not normal, which is stigmatizing.

Discussion

The authors should interpret the findings of higher PTSD among Buddhists and vocational training. Again, an explanation for each variable!

Author Response

Reviewer#3

Comments: The author examined associations between socio-demographics and PTSD in several countries. The international scope of the study is impressive.

Response: We appreciate your positive notes on our study/manuscript.

Abstract

1-Please define the first occurrence of the acronym IES-R

Response: We have defined the abbreviation ‘IES-R’ in the revised manuscript as suggested (Impact of Event Scale-Revised (IES-R). (Page-2, Line 7-8)

2-The final sentence “Religion, reason for quarantine/isolation, education level, and being cases of the infection were major predictive factors of stress during the pandemic” is a bit inaccurate; stress is different from clinical PTSD. Consider adding the list of reasons to the previous sentence or deleting e.g., “, including religion, reason. . .etc.)

Response: Thank you. We have modified this portion in the abstract as suggested to address your concerns. (Page-2, Line 20-21)

Methods

3-Why did the authors use a threshold of 24 if the PTSD diagnostic threshold is 37? They should not describe the participants as having PTSD if they do not meet diagnostic criteria. They should instead state that these individuals had elevated PTSD symptoms, or some other non-diagnostic term of their choosing.

Response: This is our mistake when the description is not really exhaustive about the diagnosis. An IES-R score of 24 or more is quite significant in the diagnosis of PTSD, however, it is only considered to have a few symptoms of PTSD or to have partial PTSD. Therefore, we have revised the term "PTSD" to "PTSDs" (PTSD symptoms) throughout the revised manuscript as appropriate. Thank you for bringing it into our notice.

4-How did the authors choose which variables to include in their regression model? Ideally there should be a quantitative, theoretical, or evidence-based reason for each variable in the model, i.e., a justification for positing an association between the variable and PTSD. Otherwise, we run into the “kitchen sink regression” problem, where numerous independent variables are selected without proper justification.

Response: We appreciate your opinion and suggestions. Based on hypotheses related to PTSDs status, we included the following variables into the optimal model selection: age, sex, religion, marital status, occupational status, level of education, salary, main laborer in family, reasons of quarantine/isolation, place of quarantine/isolation, voluntary or compulsory quarantine/isolation, environmental satisfaction quarantine/isolation, and continent. The optimal modeling method we used is Stepwise AIC on the MASS package, which uses the smallest AIC criterion to select the optimal model. By this method, we selected 6 variables related to PTSDs including: religion, level of education, reason of quarantine/ isolation, place of quarantine/isolation, voluntary or compulsory quarantine/isolation, and main laborer in the family. We beliwvw, this explanation helps to clarify the concerns raised.

5- Was the study evaluated by an institutional review board or other administrative body tasked with human subjects protection?

Response: Yes, it was. The study protocol was reviewed by IRB and this has been mentioned in methods section of the revised manuscript just before the References (Page-4, Lines 8-11). It reads as:

Ethics. The study was approved by the ethics committee of the School of Tropical Medicine and Global Health, Nagasaki University, Japan (Ref. NU_TMGH_2020_117_1). Participation was voluntary, and all participants provided informed consent on the first page of the questionnaire.

Results

  • 6- “The marital status, the average income, the reason for quarantine/isolation, the place of isolation, the voluntary to be quarantine/isolated, and the participants’ continent were significantly different between the normal and PTSD groups.(p.3, line 149)” Please specify the direction of the difference between groups for each variable . Higher, lower?

Response: Thank you. We had this data in table 1, but we didn't write the details in the text in Results section. In the revised manuscript, explanation for each variable has been added as below (Page-4, Lines 33-45):

Divorced/widowed/separated participants group had higher PTSD incidence than married/domestic partnership which in turn had higher incidence than single group.

Higher income groups reported more PTSD, with above 1000 USD group on the top, followed by 750-1000 group, while the lower income groups reported less PTSDs than non PTSD.

PTSD occurrence was also different by continents. In Africa, America, Oceania and Europe PTSD group was higher than non-PTSD, while it was reverse in Asia.

Those who were quarantined because they were confirmed to have COVID-19 (F0) reported more PTSD than non-PTSD. This trend was similar in participants who had a close contact with confirmed patients (F1), (F2) who had a close contact with people who had a direct or indirect contact with a confirmed case, (F3) who had a close contact with F2, (F4) who had a close contact with F3. While non-PTSD was greater than PTSD cases in participants who were quarantined because they lived, stayed or worked at a place nearby a confirmed COVID-19 patients group, and those who returned from affected geographical areas group.

  • 7- I would recommend using a term such as “No PTSD”, “Non-symptomatic” etc. instead of “Normal.” It implies that people with PTSD are not normal, which is stigmatizing.

Response: Thank you. We appreciate your concern and changed the term “normal” to “non-symptoms” or “non-PTSD” throughout the revised manuscript as suggested.

Discussion

8- The authors should interpret the findings of higher PTSD among Buddhists and vocational training. Again, an explanation for each variable!

Response: Thank you for your suggestion. We have added more details (including citing more references) in the discussion regarding these findings in the revised manuscript. The following text has been added in the discussion.

Our results revealed that buddhist were less likely to experience PTSDs compared to those with no religion. This accords with other previous studies that religious faith has a positive impact on the mental health and may help people to cope with emotional problems [citation:22 https://www.ncbi.nlm.nih.gov/pmc/articles/PMC7991302/ ]. Buddhism was reported to be indirectly involved with both social adaptation status and psychological well-being [citation:23  https://pubmed.ncbi.nlm.nih.gov/31549294/ ] .  Also, our study showed that individuals who possessed vocational training had a higher risk of developing PTSDs than those with higher education level (Master/Ph.D./Doctoral). This could be explained by the better perception regarding the risk of infection during quarantine or isolation period among those with higher education level, therefore, these participants had a lower risk of PTSD.

Reviewer 4 Report

Manuscript ID: ijerph-1180873

Title: Psychological Impact of COVID-19 among People under Quarantine and Isolation: A Global Survey

The findings of this study are extremely valuable for policy makers to mitigate the psychological impacts of the covid-19 pandemic, and to reduce morbidity and mortality. The study design is sound; however a revision is needed to optimize the readability and the impact of this paper.

REVIEWER’S COMMENT ON ENGLISH

Good command of English; however several paragraphs have an awkward sentence structure. Examples:

In The Introduction section

Lines 52-53 To respond the pandemic situation in the absence of definitive treatments and vaccines, health organizations and ministries employed various non-pharmacological interventions

Lines 76-77 The findings can also help to contemplate measures to provide additional support to individuals at an increased risk for adverse psychological effects of quarantine like PTSD

Change to

The findings could also inform the provision of additional support to individuals at an increased risk for adverse psychological effects of quarantine like PTSD

In the Discussion section

Line 27-28 However, our study’s findings in line with the current data regarding     the altruistic acceptance of quarantined or isolated people from previous outbreaks.

Change to

However, our study findings are in line with the current data regarding the altruistic acceptance of quarantined or isolated people from previous outbreaks.

Lines 52-53 Moreover, we did not include the questions regarding various infection-control measures implemented in the participating countries, which impeded us from commenting on the extent of their influence on the study results.

Change to:

Moreover, since we did not include questions regarding various infection-control measures implemented in the participating countries, we could not comment on the extent of their influence on the study results.

REVIEWER’S COMMENT ON TITLE

The authors might consider changing their Title to

Psychological Impacts and PTSD in People under COVID-19 Quarantine and Isolation: A Global Survey

“PTSD” is a important specifier for this study and should be included in the Title

REVIEWER’S COMMENT ON ABSTRACT

The Abstract needs to convey clearly a gap in the literature, the need for this study, measures, and the potential of the findings to inform policy changes. I suggest (below) a few edits to the Abstract to better structure its flow and to showcase the importance of the study.

Abstract

Objective: Although symptoms of posttraumatic stress disorder symptoms have been abundantly documented in quarantined/isolated individuals, healthcare workers, and survivors of the previous pandemics, including SARS and MERS), there is a paucity of evidence quantifying PTSD status globally during confinement in quarantine/isolation facilities during COVID-19. This study aimed to assess vulnerability of covid-19 quarantined/isolated people to developing PTSD. Methods: Using the Impact of Event Scale-Revised (IES-R) scale, our multicentre, multinational, cross-sectional online survey assessed the psychological impacts of the quarantine/isolation experience on 944 participants suspected or confirmed to have COVID-19, their PTSD status, and various correlates with developing PTSD. We received 2672 responses from 57 countries, with 35.33% valid responses (51.1% from females), mostly from Asian countries (635, 71.4%), of which 33.9% were from healthcare workers. Results: We identified several PTSD correlates among individuals quarantined/isolated during the pandemic. Religion, reason for quarantine/isolation, education level, and being cases of the infection were major predictive factors of stress during the pandemic. Lower rates of PTSD were observed in Buddhists religion than in participants having no religion (OR: 0.30; 95% CI: 0.13 – 0.68; p=0.005); individuals with vocational training had a higher risk of developing PTSD (OR:2.28 (1.04 – 5.15); p=0.043) compared to university graduates. Individuals forced to be quarantined/isolated had higher odds of developing PTSD than voluntarily quarantined/isolated (OR: 2.92 (1.84 – 4.74); p < 0.001). Conclusions: Understanding the vulnerability of covid-19 quarantined/isolated people to developing post-traumatic stress disorder (PTSD) can inform worldwide policies to minimize the adverse effects of such social control measures and reduce the morbidity and mortality caused by the pandemic.

I deleted “The quarantine days in the PTSD group was significantly shorter compared to the normal group (14 (range 14-40) vs. 14 (14-23.75), p=0.031)” because it is confusing and does not make a point. To the regular reader 14 is not shorter than 14.

I suggested to change from “We identified key factors” to “We identified PTSD several correlates” because correlation does not necessarily imply causation.

REVIEWER’S COMMENT on Keywords

Adequate

REVIEWER’S COMMENT on Introduction

Adequate

REVIEWER’S COMMENT on Materials and Methods

The section is described with sufficient details to allow others to replicate and build on published results.

Lines 108-109

A score of 0 to 23 was considered as normal while a score of 24 to 36 was assessed 108 as having at least a few symptoms of PTSD.

Perhaps change to:

A score of 0 to 23 was considered as normal (non-PTSD) while a score of 24 to 36 was assesses as having at least a few symptoms of PTSD.

Rationale: not having PTSD has not ruled out depression, anxiety, or other mental conditions/psychological impacts. It would be safer to name the currently “Normal group”, as the “non-PTSD group”. Then in Table 1 you will have

non-PTSD               PTSD

N=284                     N=660

Table 1

Level of education

University

What does “University” mean exactly? AA degree? BA/BA degree? University –partial education? Please add a specifier.

Table 1

Which of the following was   true   about   your

quarantine/isolation? (n=911)

I was voluntarily quarantined/isolated

Change to: “I voluntarily quarantined/isolated”

Who are you quarantined with (n=910)

No, only me

Change to: “No one else, only me”

REVIEWER’S COMMENT on Results

Appropriately divided by subheadings, this section provides a good description of the experimental results, their interpretation as well as of the conclusions drawn.

Are there any results you have not reported?

3.2 Factors associated with PTSD

“The number of quarantine days in the PTSD group was significantly shorter than that in normal group (14 (range 14-40) vs. 14 (range 14-23.75), p=0.031).

and also in Table 1.

This is confusing, please explain

REVIEWER’S COMMENT on Discussion

The authors have adequately discussed their results. A few comments:

Lines 9-10: Mak et al. reported the prevalence of PTSD in one-fourth of SARS survivors after 30 months of illness, making it the most prevalent psychiatric disorder among these individuals [21].

Shouldn’t be “30 months after the illness”?

Comfortable in isolation time

Explain why almost 75% of the PTSD group participants were comfortable with their quarantine/isolation.

In 1-2 sentences clarify for the reader that in these three subgroups

Individuals suspected of covid-19 and who isolated/quarantined voluntarily

Individuals positively diagnosed with covid-19 and who quarantined voluntarily

Individuals positively diagnosed with covid-19 and required to quarantine

the study could not determine to what extent could PTSD be attributed to the covid-19 diagnosis versus forced quarantine, versus positive diagnosis plus quarantine; the study aimed only to identify correlations.

REVIEWER’S COMMENT ON LIMITATIONS

The authors have clearly indicated the study limitations.

REVIEWER’S COMMENT ON CONCLUSIONS

It needs a bit more refinement. Below is a suggestion:

Our multicentre, multinational, cross-sectional survey has identified several key factors correlating with the development of PTSD among quarantined/isolated individuals during the COVID-19 pandemic. Religious practice, reason for being quarantined/isolated, education level, and having a positive diagnosis of infection (F0) remain predictive factors of stress during pandemics like COVID-19. These findings may inform government organizations and health ministries worldwide in drafting policies and implementing targeted measures to prevent the development of PTSD in quarantined/isolated people during a pandemic, and to reduce morbidity and mortality.

Author Response

Reviewer#4

General comment: The findings of this study are extremely valuable for policy makers to mitigate the psychological impacts of the covid-19 pandemic, and to reduce morbidity and mortality. The study design is sound; however a revision is needed to optimize the readability and the impact of this paper.

  • Response: Thank you for your positive notes on our work/ manuscript. The manuscript has been thoroughly edited/ revised to optimize the readability.

REVIEWER’S COMMENT ON ENGLISH

Good command of English; however several paragraphs have an awkward sentence structure.

Response: Thank you for pointing out the typos/ grammars. We have revised the manuscript acknowledging the suggested changes in the sentence structures, etc.

Examples:

In The Introduction section

Lines 52-53 To respond the pandemic situation in the absence of definitive treatments and vaccines, health organizations and ministries employed various non-pharmacological interventions

Response: Thank you, we have modified the sentence as suggested. Now it reads as- “In the absence of definitive treatments and vaccines, health organizations and ministries employed various non-pharmacological interventions to respond the pandemic situation” (Page-2, Lines 28-30)

Lines 76-77 The findings can also help to contemplate measures to provide additional support to individuals at an increased risk for adverse psychological effects of quarantine like PTSD

Change to

The findings could also inform the provision of additional support to individuals at an increased risk for adverse psychological effects of quarantine like PTSD.

Response: Thank you, we have modified the sentence as suggested. (Page-3, Lines 5-7)

In the Discussion section

Line 27-28 However, our study’s findings in line with the current data regarding     the altruistic acceptance of quarantined or isolated people from previous outbreaks.

Change to

However, our study findings are in line with the current data regarding the altruistic acceptance of quarantined or isolated people from previous outbreaks.

Response: Thank you, we have modified the sentence as you suggested. (Page-5, Lines 38-40)

Lines 52-53 Moreover, we did not include the questions regarding various infection-control measures implemented in the participating countries, which impeded us from commenting on the extent of their influence on the study results.

Change to:

Moreover, since we did not include questions regarding various infection-control measures implemented in the participating countries, we could not comment on the extent of their influence on the study results.

Response: Thank you, we have modified the sentence as suggested. (Page-6, Lines 11-13)

REVIEWER’S COMMENT ON TITLE

The authors might consider changing their Title to

Psychological Impacts and PTSD in People under COVID-19 Quarantine and Isolation: A Global Survey

“PTSD” is an important specifier for this study and should be included in the Title

Response: Thank you, we have added PTSD in the title as you suggested. Now it reads as- “Psychological Impacts and PTSD among People under COVID-19 Quarantine and Isolation: A Global Survey”

REVIEWER’S COMMENT ON ABSTRACT

The Abstract needs to convey clearly a gap in the literature, the need for this study, measures, and the potential of the findings to inform policy changes. I suggest (below) a few edits to the Abstract to better structure its flow and to showcase the importance of the study.

Response: Thank you, we highly appreciate your expertise, time and efforts paid to improve the abstract. We have used the edited abstract as you suggested.

Abstract

Objective: Although symptoms of posttraumatic stress disorder symptoms have been abundantly documented in quarantined/isolated individuals, healthcare workers, and survivors of the previous pandemics, including SARS and MERS), there is a paucity of evidence quantifying PTSD status globally during confinement in quarantine/isolation facilities during COVID-19. This study aimed to assess vulnerability of covid-19 quarantined/isolated people to developing PTSD. Methods: Using the Impact of Event Scale-Revised (IES-R) scale, our multicentre, multinational, cross-sectional online survey assessed the psychological impacts of the quarantine/isolation experience on 944 participants suspected or confirmed to have COVID-19, their PTSD status, and various correlates with developing PTSD. We received 2672 responses from 57 countries, with 35.33% valid responses (51.1% from females), mostly from Asian countries (635, 71.4%), of which 33.9% were from healthcare workers. Results: We identified several PTSD correlates among individuals quarantined/isolated during the pandemic. Religion, reason for quarantine/isolation, education level, and being cases of the infection were major predictive factors of stress during the pandemic. Lower rates of PTSD were observed in Buddhists religion than in participants having no religion (OR: 0.30; 95% CI: 0.13 – 0.68; p=0.005); individuals with vocational training had a higher risk of developing PTSD (OR:2.28 (1.04 – 5.15); p=0.043) compared to university graduates. Individuals forced to be quarantined/isolated had higher odds of developing PTSD than voluntarily quarantined/isolated (OR: 2.92 (1.84 – 4.74); p < 0.001). Conclusions: Understanding the vulnerability of covid-19 quarantined/isolated people to developing post-traumatic stress disorder (PTSD) can inform worldwide policies to minimize the adverse effects of such social control measures and reduce the morbidity and mortality caused by the pandemic.

Response: Thank you, we highly appreciate your expertise, time and efforts paid to improve the abstract. We have edited the abstract as accordingly to reflect your changes as far as possible. Since we have to adjust the suggestions of other reviewers too, it may not be exactly the same. Nevertheless, all your phrases, working, terminologies and suggested sentence structures have been incorporated in the abstract.

I deleted “The quarantine days in the PTSD group was significantly shorter compared to the normal group (14 (range 14-40) vs. 14 (14-23.75), p=0.031)” because it is confusing and does not make a point. To the regular reader 14 is not shorter than 14.

Response: Thank you, we have modified the text without completely deleting it.

I suggested to change from “We identified key factors” to “We identified PTSD several correlates” because correlation does not necessarily imply causation.

Response: Thank you for this reasonable change you suggested, we have modified the text accordingly in the abstract.

REVIEWER’S COMMENT on Keywords

Adequate

Response: Thank you.

REVIEWER’S COMMENT on Introduction

Adequate

Response: Thank you.

REVIEWER’S COMMENT on Materials and Methods

The section is described with sufficient details to allow others to replicate and build on published results.

Response: Thank you.

Lines 108-109

A score of 0 to 23 was considered as normal while a score of 24 to 36 was assessed 108 as having at least a few symptoms of PTSD.

Perhaps change to:

A score of 0 to 23 was considered as normal (non-PTSD) while a score of 24 to 36 was assesses as having at least a few symptoms of PTSD.

Response: Thank you, we have modified the text as you suggested. (Page-3, Line 37-38)

Rationale: not having PTSD has not ruled out depression, anxiety, or other mental conditions/psychological impacts. It would be safer to name the currently “Normal group”, as the “non-PTSD group”. Then in Table 1 you will have

non-PTSD               PTSD

N=284                     N=660

Response: Thank you, we have modified the term “normal group” to “non-PTSD” throughout the text and Tables as you suggested.

Table 1

Level of education

University

What does “University” mean exactly? AA degree? BA/BA degree? University –partial education? Please add a specifier.

Response: Thank you, we have defined the term “University” in the Table to specify and clarify.

It is in fact “undergraduate level education”. (Table-1)

Table 1

Which of the following was   true   about   your quarantine/isolation? (n=911)

I was voluntarily quarantined/isolated

Change to: “I voluntarily quarantined/isolated”

Response: Thank you for this reasonable change, we have adopted this change in the revised manuscript. (Table-1, 2)

Who are you quarantined with (n=910)

No, only me

Change to: “No one else, only me”

Response: Thank you for this reasonable change, we have adopted this change in the revised manuscript. (Table-1)

REVIEWER’S COMMENT on Results

Appropriately divided by subheadings, this section provides a good description of the experimental results, their interpretation as well as of the conclusions drawn.

Response: Thank you for your appreciation and positive notes.

Are there any results you have not reported?

Response: There is no result that is not reported in this manuscript

3.2 Factors associated with PTSD

“The number of quarantine days in the PTSD group was significantly shorter than that in normal group (14 (range 14-40) vs. 14 (range 14-23.75), p=0.031).

and also in Table 1.

This is confusing, please explain

Response: The quarantine period of subjects participating in the study varied relatively large, but in the PTSD group and the non-PTSD group, the median and 25% quartile were 14 days. But with the 75% quartile, the PTSD group was significantly higher than the non-PTSD group with 40 and 23.75 days, respectively.

REVIEWER’S COMMENT on Discussion

The authors have adequately discussed their results.

Response: Thank you for your appreciation and positive notes on our work.

A few comments:

Lines 9-10: Mak et al. reported the prevalence of PTSD in one-fourth of SARS survivors after 30 months of illness, making it the most prevalent psychiatric disorder among these individuals [21].

Shouldn’t be “30 months after the illness”?

Response: Thank you for this reasonable change, we have modified accordingly in the revised version. (page-5, Line 20-21)

Comfortable in isolation time

Explain why almost 75% of the PTSD group participants were comfortable with their quarantine/isolation.

Response: We all know that Covid-19 is transmitted directly from person to person; in the absence of vaccines, quarantine is a method that can help control spread well. Therefore, when people faced with the risk of infecting themselves and family members, most people will voluntarily comply with quarantine. Indeed, research results by Bodas (2020) in Israel showed that 94% of respondents complied with quarantine if they were compensated, and this figure was 57% if they were not compensated. Thus, this index is not much different from our study, with the voluntary quarantine rate at 75%. https://pubmed.ncbi.nlm.nih.gov/32271627/

In 1-2 sentences clarify for the reader that in these three subgroups

Individuals suspected of covid-19 and who isolated/quarantined voluntarily

Individuals positively diagnosed with covid-19 and who quarantined voluntarily

Individuals positively diagnosed with covid-19 and required to quarantine

the study could not determine to what extent could PTSD be attributed to the covid-19 diagnosis versus forced quarantine, versus positive diagnosis plus quarantine; the study aimed only to identify correlations.

Response: Thank you for your respectful suggestion. We honestly didn't structure our survey to discuss your valuable points, but we 'll use them as a new idea for a new coming survey soon.

REVIEWER’S COMMENT ON LIMITATIONS

The authors have clearly indicated the study limitations.

Response: Thank you for your positive notes on our work.

REVIEWER’S COMMENT ON CONCLUSIONS

It needs a bit more refinement. Below is a suggestion:

Our multicentre, multinational, cross-sectional survey has identified several key factors correlating with the development of PTSD among quarantined/isolated individuals during the COVID-19 pandemic. Religious practice, reason for being quarantined/isolated, education level, and having a positive diagnosis of infection (F0) remain predictive factors of stress during pandemics like COVID-19. These findings may inform government organizations and health ministries worldwide in drafting policies and implementing targeted measures to prevent the development of PTSD in quarantined/isolated people during a pandemic, and to reduce morbidity and mortality.

Response: Thank you for these reasonable changes, we have adopted the changes accordingly in the revised version as suggested. (Page-6, Line 27-34; Conclusions)

Reviewer 5 Report

Congratulations on your publication (in some similar form) because it provides a model for others. The action editor will determine the outcome of the review of your submission. This review is based on the assumption that it will be accepted in some form.

Thus, I would like to focus my review on the latter two sections of the paper, the Discussion and Conclusion.  I think that this paper has the potential to influence public policy or at least scholarly attention to public health matters. This is due to the remarkable sample of the world.

As a psychologist (human development) I am delighted that this paper and study focused on the domain of quarantine/isolation. I have read many papers, including but not limited to public health. I had wondered what it was like for those who risk PTSD during the pandemic; about the condition of the research participants. I wondered most about those diagnosed with PTSD. This interest was overshadowed by the loss in the US of a half-million souls months ago. This study provided an historical perspective by taking an important step in focusing on what their data tell them.

I believe I have reviewed more papers, applications, and funding opportunities than the norm. We have, as a body of scientists and practitioners, continued to gather together as much wisdom as possible during these days of uncertainty.  Yet the authors of this paper, as well as those of us who review it and determine it disposition, are impacted by the times we live in. In that light, I believe that this is a valuable contribution to public health research, practice, and theory. Considering the current, continuing pandemic and the welfare of the world, all of us must step up and work in collaboration to understand and manage the immediate and long-term consequences. This study provides an excellent model for understanding the impact of quarantines, isolation, and disruption of personal and family routines.

You noted what and how you conducted the study and generated the findings; that in examining the findings you found important patterns including the three factors that were especially important among mostly home quarantined and were in line with others’ findings. You noted and I applaud that your study’s findings were in line with the current data regarding the altruistic acceptance of quarantined or isolated people from previous outbreaks, for example. As you noted, further studies examining the relationship between quarantine and its psychological effects are required to reach a more definitive conclusion.

Your article was able to explain how the findings were consistent with previous ones and where different you explained why. You appropriately note in the conclusion that your findings represented a large population represent the quarantined and a good percentage of the isolated populations of the world. Your conclusions were stated directly and succinctly. You were able to identify some of the key factors contributing to the development of PTSD among your sample during the pandemic. Only three: The religiously-based reason for being quarantined, education level, and being infected remained throughout the analysis the best predictive factors of stress during pandemics.

I agree with the authors that if their findings can be replicated in other studies of quarantined populations, among others, there is justification for greater confidence in identifying those at greater risk for developing PTSD. Such a protocol would be welcomed despite the limitations of the sample.  

These findings may help government organizations worldwide in drafting policies and implementing targeted measures to reduce the development of PTSD in people throughout the world who are quarantined during a pandemic. Similar to other fields of study we need significantly greater attention to the stressors of and stress reactions to  isolation of any kind. Not only will such attention translate into decreasing the morbidity and mortality caused by the pandemic, public health programs focusing on PTSD will be more realistic and welcoming.

Author Response

Reviewer#5

Congratulations on your publication (in some similar form) because it provides a model for others. The action editor will determine the outcome of the review of your submission. This review is based on the assumption that it will be accepted in some form.

Response: Thank you for your critical review and positive notes on our work.

Thus, I would like to focus my review on the latter two sections of the paper, the Discussion and Conclusion.  I think that this paper has the potential to influence public policy or at least scholarly attention to public health matters. This is due to the remarkable sample of the world.

Response: Thank you for your critical review and positive notes on our work.

As a psychologist (human development) I am delighted that this paper and study focused on the domain of quarantine/isolation. I have read many papers, including but not limited to public health. I had wondered what it was like for those who risk PTSD during the pandemic; about the condition of the research participants. I wondered most about those diagnosed with PTSD. This interest was overshadowed by the loss in the US of a half-million souls months ago. This study provided an historical perspective by taking an important step in focusing on what their data tell them.

Response: Thank you for your critical review and positive notes on our work. This is certainly a serious issue and the concerned authority should address it based on the evidence from work (and perhaps by similar others studies too). Unfortunately, we did not follow up the people under quarantine/isolation who developed PTSD. No doubt, this should also be made a research priority.

I believe I have reviewed more papers, applications, and funding opportunities than the norm. We have, as a body of scientists and practitioners, continued to gather together as much wisdom as possible during these days of uncertainty.  Yet the authors of this paper, as well as those of us who review it and determine it disposition, are impacted by the times we live in. In that light, I believe that this is a valuable contribution to public health research, practice, and theory. Considering the current, continuing pandemic and the welfare of the world, all of us must step up and work in collaboration to understand and manage the immediate and long-term consequences. This study provides an excellent model for understanding the impact of quarantines, isolation, and disruption of personal and family routines.

Response: Thank you for your critical review and positive notes on our work, probably beyond the general horizon. Much appreciated! We do agree with you.

You noted what and how you conducted the study and generated the findings; that in examining the findings you found important patterns including the three factors that were especially important among mostly home quarantined and were in line with others’ findings. You noted and I applaud that your study’s findings were in line with the current data regarding the altruistic acceptance of quarantined or isolated people from previous outbreaks, for example. As you noted, further studies examining the relationship between quarantine and its psychological effects are required to reach a more definitive conclusion.

Response: Thank you for your critical review and positive notes on our work. We do agree with you.

Your article was able to explain how the findings were consistent with previous ones and where different you explained why. You appropriately note in the conclusion that your findings represented a large population represent the quarantined and a good percentage of the isolated populations of the world. Your conclusions were stated directly and succinctly. You were able to identify some of the key factors contributing to the development of PTSD among your sample during the pandemic. Only three: The religiously-based reason for being quarantined, education level, and being infected remained throughout the analysis the best predictive factors of stress during pandemics.

Response: Thank you for your critical review and positive notes on our work.

I agree with the authors that if their findings can be replicated in other studies of quarantined populations, among others, there is justification for greater confidence in identifying those at greater risk for developing PTSD. Such a protocol would be welcomed despite the limitations of the sample.  

Response: Thank you for your critical review and positive notes on our work. We do agree with you.

These findings may help government organizations worldwide in drafting policies and implementing targeted measures to reduce the development of PTSD in people throughout the world who are quarantined during a pandemic. Similar to other fields of study we need significantly greater attention to the stressors of and stress reactions to  isolation of any kind. Not only will such attention translate into decreasing the morbidity and mortality caused by the pandemic, public health programs focusing on PTSD will be more realistic and welcoming.

Response: Thank you for your critical review and positive notes on our work. We absolutely agree. We appreciate your time and expertise spent on our manuscript review critically.

Round 2

Reviewer 1 Report

Thank you for the revisions.

Please remove the line numbers in the title.

Author Response

Reviewer#1

Thank you for the revisions.

Please remove the line numbers in the title.

Response: Thank you for agreeing with your modifications in the revised manuscript. We have also fixed the Line-number appeared in the middle of the title. Now, there is no such Line-numbers present.